# Atom Economical Multi-Substituted Pyrrole Synthesis from Aziridine

**DOI:** 10.3390/molecules27206869

**Published:** 2022-10-13

**Authors:** Lingamurthy Macha, Ranjith Jala, Sang-Yun Na, Hyun-Joon Ha

**Affiliations:** Department of Chemistry, Hankuk University of Foreign Studies, Yongin 17035, Korea

**Keywords:** aziridine, non-activated, nucleophilic, ring-opening, regioselectivity, pyrrole

## Abstract

Multi-substituted pyrroles are synthesized from regiospecific aziridine ring-opening and subsequent intramolecular cyclization with a carbonyl group at the *γ*-position in the presence of Lewis acid or protic acid. This method is highly atom economical where all the atoms of the reactants are incorporated into the final product with the removal of water. This new protocol is applied to the synthesis of various pyrroles, including natural products.

## 1. Introduction

Pyrroles are molecules of great interest as key structural elements of various compounds, including pharmaceuticals and natural products [1,2]. For example, inonotus obliquus [3,4,5]. The white rot fungus that belongs to the family *Hymenochaetaceae* (*Basidiomycetes*) and is mainly distributed in Europe, Asia, and North America has been used for the treatment of gastrointestinal cancer, cardiovascular disease, and diabetes since the sixteenth century in Russia, Poland, and the Baltic countries. Moreover, the fungus has been reported to have anti-inflammatory [6], antioxidant [7,8,9,10], immunomodulatory [11], and hepatoprotective effects [12]. Some representative examples of 5-hydroxymethyl pyrrole-2-carbaldehydes found in the inonotus obliquus, sometimes referred to as 2-formylpyrroles or pyrralines, are displayed in Figure 1.

The synthesis of highly functionalized pyrroles has drawn considerable attention from organic and medicinal chemists. In general, the classical synthesis routes for multi-substituted pyrroles, including the Knorr condensation [13], the Paal–Knorr reaction [14], the Hantzsch reaction [15], transition metal-catalyzed reactions [16,17], and multicomponent coupling reactions [18,19,20], have been in existence for many years. However, most of them are limited by the inefficient synthesis of highly functionalized pyrroles; it is challenging to introduce various substituents to the pyrrole ring due to its harsh reaction conditions and the instability of widely used keto functionality. The construction of the pyrrole ring allows regioselective functionalization and subsequent diversification of the pyrrole ring with various substituents.

Many synthetic methods have commenced from aziridine and its derivatives by expanding the ring whose nitrogen ends at the pyrrole ring. Specifically, pyrroles are synthesized from propargyl aziridines through intramolecular cyclization and breaking of the aziridine ring with the assistance of various metal catalysts (“M”) including “Au(I)” followed by rearrangement for aromatization (Figure 1, (1)) [16,17]. Our group developed a similar pyrrole synthesis method with 3-(aziridine-2-yl)-3-hydroxypropyne taking an advantage of nucleophilic aziridine ring-opening prior to cyclization [18,19,20]. Vinyl aziridines also served as starting materials for pyrrole after 1,3-sigmatropic shift and oxidation or 2+3 cycloaddition reaction with olefin via the cleavage of the C-N bond (Figure 1, (2)). Similar [3+2]-cycloadditions were used to generate five-membered rings from 2-methyleneaziridine as a 1,3-dipole with an olefin (Figure 1, (3)). However, most of these reported methods have two critical drawbacks. First, most of the methods require a metal (“M”) catalyst. Second, only a single substituted pyrrole is generated from one set of aziridine substituents properly decorated as a starting material with the necessary counterparts, including olefins and alkynes [21,22].

In this report, we describe an atom economical synthesis of multi-substituted pyrroles from regiospecific aziridine ring-opening by various nucleophiles [23,24,25,26] and the following cyclization in Knorr-type reactions.

## 2. Results and Discussion

Treatment of hydroxy keto aziridine **1a** [25,26] with TMSN_3_ in THF or dioxane under reflux did not yield the desired pyrrole product **2a** (entries 1 and 2, Table 1). In dichloromethane, under reflux conditions, we obtained the expected pyrrole with a 70% yield (entry 3), whereas in CH_3_CN the yield increased to 85% (entry 4). In the presence of various Lewis acids such as BF_3._OEt_2_ and FeCl_3_ with NaN_3_ nucleophile, we did not obtain the desired pyrrole product **2a** (entries 5 and 6, Table 1) with all the starting materials remaining.

Next, the generality of the method was evaluated under optimized conditions that had been cyclization. This protocol provided a versatile entry for a variety of pyrroles (**2**) is well determined in Table 1. Then, we examined the scope and limitations of several *β*-(aziridin-2-yl)-*β*-hydroxy ketones (**1**) through the one-step regioselective ring-opening of aziridine followed by intramolecular to moderate yields (Figure 2).

In the successive reactions of regioselective ring opening in CH_3_CN under reflux and Knorr cyclization, the pyrrole compound **2b** was obtained in an 80% yield from the aziridine starting compounds bearing a substituent at R^2^ such as phenyl (**1b**) using TMSN_3_, whereas no pyrrole product **2c** or **2d** was obtained using TMSCl or TMSCN (see Figure 2). After TMSN_3_ screening (as mentioned in Table 1), we next screened a substrate variant using aziridines bearing a substituent at R^2^, such as *o*-methoxyphenyl (**1e**), *p*-methoxyphenyl (**1f**), and *n*-nonanyl (**1g**), as starting materials, which gave a pyrrole variant (**2e**–**2g**) in moderate to good yield under TMSN_3_ conditions. The starting substrates with an additional substituent (R^2^ as phenyl and *t*-butyldimethylsilyloxymethyl) and R^1^ as methyl and *p*-methoxyphenyl) gave pyrroles (**2h**, **2i**, and **2j**) in 75%, 72%, and 70% yields, respectively. We also applied various thiol nucleophiles under the ZnCl_2_ catalyst in MeOH to compounds (**1k**–**1m**) with substituents at C2 and C4, resulting in high yields of pyrroles (**2k**–**2m**) (Figure 2).

Next, oxidation of the secondary alcohol of compound **3** at the *γ*-position of aziridine with Dess–Martin periodinane in CH_2_Cl_2_ yielded a complex mixture of compounds, which were directly reacted for the ring-opening with various nucleophiles such as OMe, OAc, Cl, and CN to afford 2,3-disubstituted pyrrole 5-aldehydes (**4a**–**4d**) in the one-pot procedure as shown in Figure 3 with examples in the Figure 4. Whereas Swern oxidation of secondary alcohol of compound **3**, followed by regio and stereoselective aziridine ring-opening with incoming nucleophile, yielded OTBS-protected pyrrole **2** as shown in Figure 3 (see compounds **2k**–**2l** in Figure 2).

The difference in cyclization is raised by the substituent of R^2^, whether the substituent R^2^ is a simple alkyl or aryl, or hydroxymethyl in Figure 2. The initial Paal–Knorr cyclization step gives either **6** or **7**, regardless of the characteristics of R^2^, with the removal of water molecules. After the generation of the hydroxy pyrrolidine intermediate **6**, generated from most substrates with alkyl or aryl substituent on R^2^, the reaction proceeds to aromatization to yield **2** as shown in Figure 2. From the substrate-bearing hydroxymethyl group, the ammonium ion intermediate **8** was generated, from which the deprotonation occurs to give **9** and its resonance form as **10**. One more deprotonation from **10** gives rise to the final 2-formyl pyrroles **4**, as shown in Figure 4 (Figure 5).

This method was extended to the synthesis of the natural product inotopyrrole **19** (Figure 5). Treatment of compound **11** with Weinreb salt and *i*-PrMgCl to give compound **12**, followed by allyl magnesium bromide and a subsequent reduction of aziridine ketone by NaBH_4_ yielded the alcohol compound **13** in 68% yield for two steps. Protection of the secondary alcohol with TBSOTf and 2,6-lutidine to furnish olefin **14** at a 73% yield. Olefin **14** was subjected to simple dihydroxylation using OsO_4_ and NMO to give a diol compound, followed by selective protection of the primary alcohol with TBSCl to afford secondary alcohol, and subsequently, Swern oxidation of alcohol afforded key intermediate keto compound **15** in a 62% yield. Then, we applied our optimized method on compound **15** for the synthesis of pyrrole derivative **16** from a one-step regioselective ring-opening followed by cyclization of keto compound by using AcOH and CH_2_Cl_2_ at 0 °C in 82% yield. Then, deacetylation of **16** with K_2_CO_3_ to give alcohol **17**, followed by Dess-martin oxidation of primary alcohol, afforded aldehyde **18** with a 74% yield. Removal of the TBS group with TBAF gave rise to the desired natural product, inotopyrrole (**19**), in an 84% yield. Spectral data (^1^H, ^13^C NMR) and HRMS data of our synthetic ionotopyrrole (**19**) were in full agreement with those reported for the natural product (Figure 6) [3,4,5].

## 3. Materials and Methods

### 3.1. General Information

Chiral aziridines are available from Sigma-Aldrich as reagents. They are also available from Imagene Co., Ltd. (http://www.imagene.co.kr/) in bulk quantities. All commercially available compounds were used as received unless stated otherwise. All reactions were carried out under an atmosphere of nitrogen in oven-dried glassware with magnetic stirrer. Dichloromethane was distilled from calcium hydride. Reactions were monitored by thin layer chromatography (TLC) with 0.25 mm E. Merck pre-coated silica gel plates (60 F254). Visualization was accomplished with either UV light, or by immersion in solutions of ninhydrin, *p*-anisaldehyde, or phosphomolybdic acid (PMA) followed by heating on a hot plate for about 10 sec. Purification of reaction products was carried out by flash chromatography using Kieselgel 60 Art 9385 (230–400 mesh). The ^1^H-NMR and ^13^C-NMR spectra were obtained using Varian unity lNOVA 400WB (400 MHz) or Bruker AVANCE III HD (400 MHz) spectrometer. Chemical shifts are reported relative to chloroform (δ = 7.26) for ^1^H NMR, chloroform (δ = 77.2) for ^13^C NMR, acetonitrile (δ = 1.94) for ^1^H NMR, and acetonitrile (δ = 1.32) for ^13^C NMR (see Appendix A). Data are reported as br = broad, s = singlet, d = doublet, t = triplet, q = quartet, p = quintet, m = multiplet. Coupling constants are given in Hz. Ambiguous assignments were resolved using standard one-dimensional proton decoupling experiments. Optical rotations were obtained using a Rudolph Autopol III digital polarimeter and JASCO P-2000. Optical rotation data are reported as follows: [α]^20^ (concentration c = g/100 mL, solvent). High-resolution mass spectra were recorded on a 4.7 Tesla IonSpec ESI-TOFMS, JEOL (JMS-700), and an AB Sciex 4800 Plus MALDI TOF^TM^, (2,5-dihydroxybenzoic acid (DHB) matrix was used to prepare samples for MS. Data were obtained in the reflector positive mode with internal standards for calibration).

### 3.2. General Procedure for the Synthesis of Pyrroles

To a stirred solution of **1a** (100 mg, 0.38 mmol) in CH_3_CN (3 mL) was added trimethylsilyl azide (0.1 mL, 0.76 mmol) at 90 °C. After being stirred for 4 h, the mixture was concentrated under reduced pressure. The crude product was purified by column chromatography (EtOAc/hexane = 1:9) to afford pyrrole compound **2a**.

#### (*R*)-2-(Azidomethyl)-1-(1-phenylethyl)-5-propyl-1*H*-pyrrole (**2a**)

Yellow liquid, (80 mg) 85% yield. The ^1^H NMR (400 MHz, CDCl_3_) δ 7.32 (ddd, *J* = 7.6, 5.0, 2.0 Hz, 3H), 7.07–7.02 (m, 2H), 6.17 (d, *J* = 3.5 Hz, 1H), 5.92 (d, *J* = 3.5 Hz, 1H), 5.53 (q, *J* = 7.2 Hz, 1H), 4.17 (d, *J* = 14.5 Hz, 1H), 3.93 (d, *J* = 14.4 Hz, 1H), 2.49–2.41 (m, 1H), 2.35–2.25 (m, 1H), 1.90 (d, *J* = 7.2 Hz, 3H), 1.61–1.50 (m, 2H), 0.88 (t, *J* = 7.3 Hz, 3H). The ^13^C NMR (101 MHz, CDCl_3_) δ 141.9, 136.1, 128.5, 127.1, 125.9, 124.7, 110.9, 105.6, 52.6, 47.6, 29.6, 22.1, 19.6, 14.0. HRMS-ESI (*m*/*z*): [M + H]^+^ calcd. for C_16_H_21_N_4_, 269.6121, found 269.6128. Copies of ^1^H and ^13^C NMR could be found in Appendix A.

#### 2-(Azidomethyl)-1-benzyl-5-phenyl-1*H*-pyrrole (**2b**)

Yellow liquid, (90 mg) 80% yield. The ^1^H NMR (400 MHz, CDCl_3_) δ 7.28 (ddd, *J* = 10.9, 6.7, 3.5 Hz, 8H), 6.89 (d, *J* = 7.2 Hz, 2H), 6.33 (d, *J* = 3.6 Hz, 1H), 6.25 (d, *J* = 3.6 Hz, 1H), 5.21 (s, 2H), 4.15 (s, 2H). The ^13^C NMR (101 MHz, CDCl_3_) δ 138.6, 137.3, 133.0, 128.9, 128.8, 128.5, 127.4, 127.3, 127.0, 125.5, 111.3, 108.4, 47.7, 47.2. HRMS-ESI (*m*/*z*): [M + H]^+^ calcd. for C_18_H_17_N_4_, 289.1358, found 289.1362. Copies of ^1^H and ^13^C NMR could be found in Appendix A.

#### 2-(Azidomethyl)-1-benzyl-5-(2-methoxyphenyl)-1*H*-pyrrole (**2e**)

Yellow liquid, (93 mg) 78% yield. The ^1^H NMR (400 MHz, CDCl_3_) δ 7.39–7.03 (m, 5H), 7.00–6.78 (m, 4H), 6.34 (d, *J* = 3.5 Hz, 1H), 6.16 (d, *J* = 3.5 Hz, 1H), 5.03 (s, 2H), 4.14 (s, 2H), 3.65 (s, 3H). The ^13^C NMR (101 MHz, CDCl_3_) δ 157.4, 138.6, 133.6, 132.7, 129.6, 128.4, 127.0, 126.4, 126.0, 122.1, 120.6, 111.0, 110.8, 108.6, 55.3, 48.2. HRMS-ESI (*m*/*z*): [M + H]^+^ calcd. for C_19_H_19_N_4_O, 319.0446, found 319.0449. Copies of ^1^H and ^13^C NMR could be found in Appendix A.

#### 2-(Azidomethyl)-1-benzyl-5-(4-methoxyphenyl)-1*H*-pyrrole (**2f**)

Yellow liquid, (89 mg) 85% yield. The ^1^H NMR (400 MHz, CDCl_3_) δ 7.01 (ddd, *J* = 6.6, 5.2, 2.7 Hz, 5H), 6.67–6.65 (m, 2H), 6.62–6.59 (m, 2H), 6.09 (d, *J* = 3.5 Hz, 1H), 5.96 (d, *J* = 3.5 Hz, 1H), 4.95 (s, 2H), 3.89 (s, 2H), 3.54 (s, 3H). The ^13^C NMR (101 MHz, CDCl_3_) δ 159.1, 138.7, 137.0, 130.3, 128.8, 127.2, 126.4, 125.5, 125.5, 113.9, 111.1, 107.8, 55.3, 47.6, 47.3. HRMS-ESI (*m*/*z*): [M + H]^+^ calcd. for C_19_H_19_N_4_O, 319.1228, found 319.1230. Copies of ^1^H and ^13^C NMR could be found in Appendix A.

#### 2-(Azidomethyl)-1-benzyl-5-nonyl-1*H*-pyrrole (**2g**)

Yellow liquid, (85 mg) 82% yield. The ^1^H NMR (400 MHz, CDCl_3_) δ 7.37–7.28 (m, 3H), 6.92 (d, *J* = 7.0 Hz, 2H), 6.27 (d, *J* = 3.5 Hz, 1H), 6.01 (d, *J* = 3.5 Hz, 1H), 5.18 (s, 2H), 4.20 (s, 2H), 2.51 (dd, *J* = 13.6, 6.0 Hz, 2H), 1.67–1.59 (m, 2H), 1.38–1.29 (m, 12H), 0.94 (t, *J* = 6.9 Hz, 3H). The ^13^C NMR (101 MHz, CDCl_3_) δ 138.3, 136.2, 128.7, 127.2, 125.5, 125.4, 125.0, 110.2, 105.3, 47.2, 46.9, 31.8, 29.3, 28.6, 26.5, 22.5, 13.8. HRMS-ESI (*m*/*z*): [M + H]^+^ calcd. for C_21_H_31_N_4_, 339.4618, found 339.4620. Copies of ^1^H and ^13^C NMR could be found in Appendix A.

#### (*R*)-2-(Azidomethyl)-3-methyl-5-phenyl-1-(1-phenylethyl)-1*H*-pyrrole (**2h**)

Yellow liquid, (83 mg) 75% yield. The ^1^H NMR (400 MHz, CDCl_3_) δ 7.35–7.24 (m, 8H), 7.04–7.02 (m, 2H), 6.08 (s, 1H), 5.59 (q, *J* = 7.3 Hz, 1H), 4.23 (d, *J* = 14.7 Hz, 1H), 3.75 (d, *J* = 14.8 Hz, 1H), 2.16 (s, 3H), 1.88 (d, *J* = 7.2 Hz, 3H). The ^13^C NMR (101 MHz, CDCl_3_) δ 142.4, 133.8, 129.4, 128.6, 128.4, 127.4, 127.1, 125.8, 122.6, 121.7, 110.3, 53.2, 45.1, 19.9, 11.3. HRMS-ESI (*m*/*z*): [M + H]^+^ calcd. for C_20_H_21_N_4_, 317.5973, found 317.5975. Copies of ^1^H and ^13^C NMR could be found in Appendix A.

#### (*R*)-2-(Azidomethyl)-5-(((*tert*-butyldimethylsilyl)oxy)methyl)-3-methyl-1-(1-phenylethyl)-1*H*-pyrrole (**2i**)

Yellow liquid, (91 mg) 72% yield. The ^1^H NMR (400 MHz, CDCl_3_) δ 7.34–7.26 (m, 3H), 7.18–7.14 (m, 2H), 5.97 (s, 1H), 5.72 (q, *J* = 7.2 Hz, 1H), 4.53 (s, 2H), 4.22 (d, *J* = 14.7 Hz, 1H), 3.81 (d, *J* = 14.7 Hz, 1H), 2.12 (s, 3H), 1.94 (d, *J* = 7.2 Hz, 3H), 0.88 (s, 9H), 0.05 (d, *J* = 7.1 Hz, 6H). The ^13^C NMR (101 MHz, CDCl_3_) δ 142.0, 133.0, 128.4, 127.1, 126.2, 122.6, 119.9, 109.7, 57.8, 53.1, 44.6, 25.8, 19.6, 18.2, 11.2, −5.2. HRMS-ESI (*m*/*z*): [M + Na]^+^ calcd. for C_21_H_32_N_4_NaOSi, 407.8471, found 407.8474. Copies of ^1^H and ^13^C NMR could be found in Appendix A.

#### (*R*)-2-(Azidomethyl)-5-(((*tert*-butyldimethylsilyl)oxy)methyl)-3-(4-methoxyphenyl)-1-(1-phenylethyl)-1*H*-pyrrole (**2j**)

Yellow liquid, (87 mg) 70% yield. The ^1^H NMR (400 MHz, CDCl_3_) δ 7.40–7.34 (m, 4H), 7.32–7.29 (m, 1H), 7.22 (dd, *J* = 5.1, 4.2 Hz, 2H), 6.97 (d, *J* = 8.8 Hz, 2H), 6.25 (s, 1H), 5.82 (q, *J* = 7.1 Hz, 1H), 4.55 (s, 2H), 4.34 (d, *J* = 14.6 Hz, 1H), 4.05 (d, *J* = 14.6 Hz, 1H), 3.87 (s, 3H), 2.04 (d, *J* = 7.2 Hz, 3H), 0.91 (s, 9H), 0.08 (d, *J* = 3.9 Hz, 6H). The ^13^C NMR (101 MHz, CDCl_3_) δ 158.2, 141.6, 133.7, 129.6, 128.6, 128.5, 127.3, 126.4, 126.3, 122.5, 113.9, 109.1, 57.9, 55.3, 53.6, 45.4, 25.9, 19.6, 18.2, −5.2. HRMS-ESI (*m*/*z*): [M + H]^+^ calcd. for C_27_H_37_N_4_O_2_Si, 477.0417, found 477.0419. Copies of ^1^H and ^13^C NMR could be found in Appendix A.

#### (*R*)-5-(((5-(((*tert*-Butyldimethylsilyl)oxy)methyl)-3-methyl-1-(1-phenylethyl)-1*H*-pyrrol-2-yl)methyl)thio)-1-phenyl-1*H*-tetrazole (**2k**)

Yellow liquid, (107 mg) 82% yield. The ^1^H NMR (400 MHz, CDCl_3_) δ 7.83–7.80 (m, 2H), 7.56–7.48 (m, 3H), 7.21 (t, *J* = 7.6 Hz, 2H), 7.11–7.03 (m, 3H), 6.08 (s, 1H), 5.85 (q, *J* = 7.1 Hz, 1H), 5.28 (s, 2H), 4.53 (s, 2H), 2.19 (s, 3H), 1.93 (d, *J* = 7.2 Hz, 3H), 0.88 (s, 9H), 0.06 (d, *J* = 2.8 Hz, 6H). The ^13^C NMR (101 MHz, CDCl_3_) δ 162.1, 141.3, 134.7, 133.6, 129.4, 129.1, 128.2, 126.8, 126.1, 123.6, 121.4, 120.3, 110.5, 58.0, 53.5, 42.6, 25.9, 19.9, 18.3, 11.5, −5.2. HRMS-ESI (*m*/*z*): [M + H]^+^ calcd. for C_28_H_38_N_5_OSSi, 520.4336, found 520.4340. Copies of ^1^H and ^13^C NMR could be found in Appendix A.

#### 5-(((1-Benzyl-5-nonyl-1*H*-pyrrol-2-yl)methyl)thio)-1-phenyl-1*H*-tetrazole (**2l**)

Yellow liquid, (115 mg) 81% yield. The ^1^H NMR (400 MHz, CDCl_3_) δ 7.66 (dd, *J* = 8.2, 1.5 Hz, 2H), 7.51–7.43 (m, 3H), 7.17 (t, *J* = 7.4 Hz, 2H), 7.11 (d, *J* = 7.3 Hz, 1H), 6.67 (d, *J* = 7.5 Hz, 2H), 6.52 (d, *J* = 3.5 Hz, 1H), 6.03 (d, *J* = 3.5 Hz, 1H), 5.44 (s, 2H), 5.32 (s, 2H), 2.44–2.39 (m, 2H), 1.57 (dd, *J* = 15.0, 7.4 Hz, 2H), 1.30–1.22 (m, 12H), 0.87 (t, *J* = 6.8 Hz, 3H). The ^13^C NMR (101 MHz, CDCl_3_) δ 162.4, 137.9, 136.5, 134.5, 129.4, 128.9, 128.4, 127.0, 124.9, 123.8, 123.4, 111.9, 105.7, 46.9, 43.6, 31.8, 29.5, 29.4, 29.3, 28.5, 26.4, 22.6, 14.1. HRMS-ESI (*m*/*z*): [M + H]^+^ calcd. for C_28_H_36_N_5_S, 474.3226, found 474.3228. Copies of ^1^H and ^13^C NMR could be found in Appendix A.

#### 1-Benzyl-2-(((4-methoxybenzyl)thio)methyl)-5-phenyl-1*H*-pyrrole (**2m**)

Yellow liquid, (105 mg) 75% yield. The ^1^H NMR (400 MHz, CDCl_3_) δ 7.32–7.29 (m, 4H), 7.25–7.15 (m, 6H), 6.86–6.79 (m, 4H), 6.22 (d, *J* = 3.5 Hz, 1H), 6.17 (d, *J* = 3.5 Hz, 1H), 5.25 (s, 2H), 3.79 (s, 3H), 3.62 (s, 2H), 3.44 (s, 2H). The ^13^C NMR (101 MHz, CDCl_3_) δ 158.5, 138.9, 136.2, 133.4, 130.3, 130.0, 128.8, 128.6, 128.3, 126.9, 125.6, 113.8, 110.0, 108.0, 55.3, 47.4, 34.9, 27.5. HRMS-ESI (*m*/*z*): [M + Na]^+^ calcd. for C_26_H_25_NNaOS, 422.5371, found 422.5375. Copies of ^1^H and ^13^C NMR could be found in Appendix A.

#### (*R*)-5-(methoxymethyl)-4-methyl-1-(1-phenylethyl)-1*H*-pyrrole-2-carbaldehyde (**4a**)

To a stirred solution of secondary alcohol **3** (200 mg, 0.527 mmol) was dissolved in CH_2_Cl_2_ (6 mL) under N_2_ at 0 °C and Dess–Martin periodinane (335 mg, 0.791 mmol) was added to the reaction mixture and allowed to stir for 2 h. Ether was added to the reaction mixture and the solid was filtered. The filtrate was washed with saturated NaHCO_3_ solution, dried over anhydrous Na_2_SO_4_, and solvents were removed under vacuum to obtain a crude product, which was used for the next reaction without further purification.

To a stirred solution of above crude ketone compound was dissolved in MeOH (3 mL) under N_2_ at 0 °C and ZnCl_2_ (86 mg, 0.632 mmol) was added to the reaction mixture and allowed to stir for 2 h. After 2 h, the reaction mixture was diluted with CH_2_Cl_2_ (10 mL), quenched with water, and extracted with CH_2_Cl_2_ (2 × 10 mL). The organic layer was dried over Na_2_SO_4_ and concentrated in vacuo to obtain a crude product, which was purified by silica gel column chromatography (EtOAc/hexane, 1:9) to obtain pyrrole compound **4a** (102 mg, 75% yield) as a yellow liquid. The ^1^H NMR (400 MHz, CDCl_3_) δ 9.41 (s, 1H), 7.32–7.26 (m, 3H), 7.13 (d, *J* = 8.1 Hz, 2H), 6.80 (s, 1H), 6.56 (s, 1H), 4.20 (d, *J* = 12.4 Hz, 1H), 4.07 (d, *J* = 12.4 Hz, 1H), 3.21 (s, 3H), 2.12 (s, 3H), 1.91 (d, *J* = 7.1 Hz, 3H). The ^13^C NMR (101 MHz, CDCl_3_) δ 179.1, 141.5, 136.3, 131.4, 128.3, 127.0, 126.1, 125.1, 121.7, 63.5, 57.7, 53.9, 19.4, 11.1. HRMS-ESI (*m*/*z*): [M + H]^+^ calcd. for C_16_H_20_NO_2_, 258.2714, found 258.2718. Copies of ^1^H and ^13^C NMR could be found in Appendix A.

#### (*R*)-5-(Chloromethyl)-4-methyl-1-(1-phenylethyl)-1*H*-pyrrole-2-carbaldehyde (**4b**)

Yellow liquid, (92 mg) 70% yield. The ^1^H NMR (400 MHz, CDCl_3_) δ 9.50 (s, 1H), 7.37–7.29 (m, 3H), 7.16–7.13 (m, 2H), 6.82 (s, 1H), 6.77 (s, 1H), 4.43 (d, *J* = 12.8 Hz, 1H), 4.27 (d, *J* = 12.9 Hz, 1H), 2.15 (s, 3H), 2.00 (d, *J* = 7.2 Hz, 3H). The ^13^C NMR (101 MHz, CDCl_3_) δ 179.6, 141.1, 135.3, 131.6, 128.6, 128.4, 127.3, 125.9, 125.3, 53.8, 35.4, 19.4, 10.8. HRMS-ESI (*m*/*z*): [M + H]^+^ calcd. for C_15_H_17_ClNO, 262.1479, found 262.1483. Copies of ^1^H and ^13^C NMR could be found in Appendix A.

#### (*R*)-(5-Formyl-3-methyl-1-(1-phenylethyl)-1*H*-pyrrol-2-yl)methyl acetate (**4c**)

Yellow liquid, (98 mg) 78% yield. The ^1^H NMR (400 MHz, CDCl_3_) δ 9.47 (s, 1H), 7.32–7.26 (m, 3H), 7.11 (d, *J* = 8.0 Hz, 2H), 6.81 (s, 1H), 6.65 (s, 1H), 4.96 (d, *J* = 13.4 Hz, 1H), 4.69 (d, *J* = 13.4 Hz, 1H), 2.11 (s, 3H), 1.91 (t, *J* = 3.5 Hz, 6H). The ^13^C NMR (101 MHz, CDCl_3_) δ 179.5, 170.3, 141.0, 134.0, 131.8, 128.4, 127.2, 126.0, 125.4, 55.6, 53.9, 20.6, 19.4, 10.9. HRMS-ESI (*m*/*z*): [M + H]^+^ calcd. for C_17_H_20_NO_3_, 286.6442, found 286.6446. Copies of ^1^H and ^13^C NMR could be found in Appendix A.

#### (*R*)-2-(5-Formyl-3-methyl-1-(1-phenylethyl)-1*H*-pyrrol-2-yl)acetonitrile (**4d**)

Yellow liquid, (93 mg) 65% yield. The ^1^H NMR (400 MHz, CDCl_3_) δ 9.47 (s, 1H), 7.32–7.27 (m, 3H), 7.12–7.09 (m, 2H), 6.81 (s, 1H), 6.65 (s, 1H), 4.96 (d, *J* = 13.4 Hz, 1H), 4.69 (d, *J* = 13.3 Hz, 1H), 2.11 (s, 3H), 1.90 (d, *J* = 2.2 Hz, 3H). The ^13^C NMR (101 MHz, CDCl_3_) δ 179.4, 141.0, 133.9, 131.8, 128.5, 128.4, 127.2, 126.0, 125.9, 125.3, 55.6, 53.9, 20.6, 19.4, 10.9. HRMS-ESI (*m*/*z*): [M + H]^+^ calcd. for C_16_H_17_N_2_O, 253.4441, found 253.4446. Copies of ^1^H and ^13^C NMR could be found in Appendix A.

#### Ethyl 1-phenethylaziridine-2-carboxylate (**11**)

To a stirred solution of ethyl 2,3-dibromopropanoate (5.0 g, 19.30 mmol, 1.0 equiv) dissolved in acetonitrile (60 mL), were added potassium carbonate (8.0 g, 57.9 mmol, 3.0 equiv) followed by 2-phenylethanamine (2.9 mL, 23.16 mmol, 1.2 equiv) in dropwise manner at room temperature and reaction mixture were allowed to stir for 12 h. After completion, quenched with water (25 mL) and filtered out over filter paper (pore size 8–10 µm). The organic mixture was extracted with Et_2_O (2 × 30 mL), dried over anhydrous magnesium sulfate, and concentrated under reduced pressure to obtain a crude mixture of Ethyl 1-phenethylaziridine-2-carboxylate **11** as a yellow liquid (3.8 g, 89%). The ^1^H NMR (400 MHz, CDCl_3_) δ 7.27 (ddd, *J* = 7.4, 3.1, 1.3 Hz, 2H), 7.22–7.16 (m, 3H), 4.24–4.11 (m, 2H), 2.93 (dd, *J* = 15.1, 6.9 Hz, 2H), 2.65–2.49 (m, 2H), 2.14 (dd, *J* = 3.1, 1.2 Hz, 1H), 1.94 (dd, *J* = 6.5, 3.1 Hz, 1H), 1.52 (dd, *J* = 6.5, 1.1 Hz, 1H), 1.27 (t, *J* = 7.1 Hz, 3H). The ^13^C NMR (101 MHz, CDCl_3_) δ 170.7, 139.3, 128.7, 128.3, 126.1, 62.3, 61.0, 37.5, 36.0, 34.3, 14.1. HRMS-ESI (*m*/*z*): [M + H]^+^ calcd. for C_13_H_18_NO_2_, 220.6121, found 220.6128. Copies of ^1^H and ^13^C NMR could be found in Appendix A.

#### *N*-Methoxy-*N*-methyl-1-phenethylaziridine-2-carboxamide (**12**)

To a stirred solution of ester **11** (3.8 g, 17.35 mmol) and *N*,*O*-dimethylhydroxylamine hydrochloride (2.53 g, 26.0 mmol) in dry THF (50 mL) at 0 °C was slowly added *i*-PrMgCl (26.0 mL, 2.0 M in THF, 52.05 mmol), and the reaction mixture was stirred for 1 h. The reaction mixture was quenched with saturated NH_4_Cl solution and extracted with EtOAc (3 × 20 mL). The combined organic layers were dried over anhydrous Na_2_SO_4_ and concentrated in vacuo to obtain the crude product, which was purified by silica gel column chromatography (EtOAc/hexanes, 1:1) to afford Weinreb amide **12** as a yellow color oil (3.2 g, 78.8%) yield. The ^1^H NMR (400 MHz, CDCl_3_) δ 7.29–7.25 (m, 2H), 7.20 (d, *J* = 7.2 Hz, 3H), 3.68 (s, 3H), 3.21 (s, 3H), 2.99–2.88 (m, 2H), 2.71 (ddd, *J* = 11.4, 8.7, 6.6 Hz, 1H), 2.56–2.42 (m, 2H), 2.17 (dd, *J* = 3.2, 1.3 Hz, 1H), 1.51 (dd, *J* = 6.5, 1.2 Hz, 1H). The ^13^C NMR (101 MHz, CDCl_3_) δ 170.3, 139.6, 128.7, 128.3, 126.1, 62.7, 61.6, 36.1, 35.3, 34.0, 32.5. HRMS-ESI (*m*/*z*): [M + H]^+^ calcd. for C_13_H_19_N_2_O_2_, 235.0336, found 234.0340. Copies of ^1^H and ^13^C NMR could be found in Appendix A.

#### 1-(1-Phenethylaziridin-2-yl)but-3-en-1-ol (**13**)

To a stirred solution of Weinreb amide **12** (3.2 g, 13.67 mmol) was slowly added allylMgBr (8.2 mL, 2.0 M in THF, 16.4 mmol) in dry THF (40 mL) at 0 °C, and the reaction mixture was stirred for 1 h. The reaction mixture was quenched with saturated NH_4_Cl solution and extracted with EtOAc (2 × 20 mL). The combined organic layers were dried over anhydrous Na_2_SO_4_ and concentrated in vacuo to obtain the crude allyl product, which was used for the next reaction without further purification.

To a stirred solution of above keto compound (3.2 g, 14.86 mmol) was slowly added NaBH_4_ (0.45 g, 11.88 mmol) in MeOH (40 mL) at 0 °C, and the reaction mixture was stirred for 1 h. Then, MeOH was removed under vacuum and extracted with CH_2_Cl_2_ (2 × 10 mL). The organic layer was dried over Na_2_SO_4_ and concentrated in vacuo to obtain the crude allyl alcohol product, which was purified by column chromatography (EtOAc/hexanes, 2:8) to give pure 1-(1-phenethylaziridin-2-yl)but-3-en-1-ol (**13**) as a yellow liquid (2.6 g, 87%) yield. The ^1^H NMR (400 MHz, CDCl_3_) δ 7.33–7.16 (m, 5H), 5.84 (ddt, *J* = 17.2, 10.2, 7.1 Hz, 1H), 5.16–5.06 (m, 2H), 3.66 (td, *J* = 6.3, 3.8 Hz, 1H), 2.85 (t, *J* = 7.4 Hz, 2H), 2.67 (dt, *J* = 11.6, 7.3 Hz, 1H), 2.52–2.43 (m, 1H), 2.24 (t, *J* = 6.7 Hz, 2H), 1.80 (d, *J* = 3.6 Hz, 1H), 1.49 (dt, *J* = 7.0, 3.7 Hz, 1H), 1.23 (d, *J* = 6.4 Hz, 1H). The ^13^C NMR (101 MHz, CDCl_3_) δ 139.7, 134.3, 128.7, 128.3, 126.1, 117.4, 67.9, 61.7, 42.2, 39.3, 36.3, 29.3. HRMS-ESI (*m*/*z*): [M + H]^+^ calcd. for C_14_H_20_NO, 218.0231, found 218.0234. Copies of ^1^H and ^13^C NMR could be found in Appendix A.

#### 2-(1-((*tert*-Butyldimethylsilyl)oxy)but-3-en-1-yl)-1-phenethylaziridine (**14**)

To a stirred solution of allyl alcohol **13** (2.5 g, 11.50 mmol) in dry CH_2_Cl_2_ (30 mL) was added imidazole (1.5 g, 23.0 mmol) and TBSCl (1.9 g, 12.65 mmol), sequentially, at 0 °C under an N_2_ atmosphere. After 4 h of being stirred at rt, the reaction mixture was quenched with saturated aqueous NH_4_Cl (10 mL). The organic layer was separated, and the aqueous layer was extracted with CH_2_Cl_2_ (2 × 20 mL). The organic layer was dried over Na_2_SO_4_ and concentrated in vacuo to obtain the crude product, which was purified by column chromatography (EtOAc/hexanes, 2:8) to give pure 2-(1-((*tert*-butyldimethylsilyl)oxy)but-3-en-1-yl)-1-phenethylaziridine **14** as a yellow liquid (2.8 g, 73%) yield. The ^1^H NMR (400 MHz, CDCl_3_) δ 7.30–7.24 (m, 2H), 7.19 (dd, *J* = 7.1, 5.2 Hz, 3H), 5.91 (ddt, *J* = 17.1, 10.2, 7.1 Hz, 1H), 5.13–5.04 (m, 2H), 3.20 (td, *J* = 7.0, 4.4 Hz, 1H), 2.86 (t, *J* = 8.0 Hz, 2H), 2.55 (dt, *J* = 11.5, 7.7 Hz, 1H), 2.48–2.33 (m, 3H), 1.69 (d, *J* = 3.4 Hz, 1H), 1.45 (ddd, *J* = 7.6, 6.4, 3.4 Hz, 1H), 1.29 (d, *J* = 6.3 Hz, 1H), 0.88 (s, 9H), 0.02 (d, *J* = 2.1 Hz, 6H). The ^13^C NMR (101 MHz, CDCl_3_) δ 139.8, 135.0, 128.6, 128.3, 126.0, 116.9, 74.6, 62.7, 43.6, 40.9, 36.3, 33.9, 25.8, 18.1, −4.1, −4.6. HRMS-ESI (*m*/*z*): [M + H]^+^ calcd. for C_20_H_34_NOSi, 332.1222, found 332.1224. Copies of ^1^H and ^13^C NMR could be found in Appendix A.

#### Octamethyl-8-(1-phenethylaziridin-2-yl)-4,9-dioxa-3,10-disiladodecan-6-one (**15**)

To a stirred solution of 2-(1-((*tert*-butyldimethylsilyl)oxy)but-3-en-1-yl)-1-phenethylaziridine **14** (2.5 g, 7.5 mmol) and *N*-Methylmorpholine N-oxide (2.64 g, 22.61 mmol) in acetone: H_2_O (4:1) (20 mL) at room temperature was slowly added OsO_4_ (3.2 mL, 0.75 mmol), and the reaction mixture was stirred for 6 h. The reaction mixture was quenched with saturated NH_2_SO_3_ solution and extracted with EtOAc (3 × 20 mL). The combined organic layers were dried over anhydrous Na_2_SO_4_ and concentrated in vacuo to obtain the crude dihydroxy product, which was used for the next reaction without further purification.

To a stirred solution of dihydroxy alcohol (2.5 g, 6.8 mmol) in dry CH_2_Cl_2_ (30 mL) was added imidazole (0.93 g, 13.67 mmol) and TBSCl (1.13 g, 7.5 mmol), sequentially, at 0 °C under an N_2_ atmosphere. After 2 h of being stirred at rt, the reaction mixture was quenched with saturated aqueous NH_4_Cl (10 mL). The organic layer was separated, and the aqueous layer was extracted with CH_2_Cl_2_ (2 × 20 mL). The organic layer was dried over Na_2_SO_4_ and concentrated in vacuo to obtain the crude product, which was used for the next reaction without further purification.

To a solution of oxalyl chloride (0.67 mL, 7.81 mmol) in CH_2_Cl_2_ (20 mL) at −78 °C was added dimethyl sulfoxide (1.1 mL, 15.63 mmol) over 15 min. The resulting mixture was stirred for another 45 min and then a solution of alcohol (2.5 g, 5.21 mmol) in CH_2_Cl_2_ (20 mL) was added dropwise. The resulting white suspension was stirred for 2h before adding triethylamine (2.18 mL, 15.63 mmol). The reaction mixture was stirred for 30 min at −78 °C and then warmed to 0 °C and allowed to stir for 15 min. The reaction mixture was quenched with water (20 mL) and the aqueous phase was extracted with CH_2_Cl_2_ (2 × 20 mL). The combined organic layers were washed with brine, dried over anhydrous Na_2_SO_4_, and concentrated under reduced pressure to obtain a crude, which was purified by column chromatography (EtOAc/hexanes, 2:8) to give pure Octamethyl-8-(1-phenethylaziridin-2-yl)-4,9-dioxa-3,10-disiladodecan-6-one **15** as a yellow liquid (2.1 g, 62%) yield. The ^1^H NMR (400 MHz, CDCl_3_) δ 7.27–7.25 (m, 2H), 7.18 (t, *J* = 7.6 Hz, 3H), 4.17 (s, 2H), 4.00–3.90 (m, 1H), 2.82 (t, *J* = 6.9 Hz, 2H), 2.65–2.59 (m, 2H), 2.59–2.52 (m, 1H), 2.35–2.28 (m, 1H), 1.66 (d, *J* = 2.5 Hz, 1H), 1.54 (dd, *J* = 9.0, 6.4 Hz, 1H), 1.18 (d, *J* = 6.0 Hz, 1H), 0.92 (s, 9H), 0.87 (s, 9H), 0.10 (s, 6H), 0.08 (s, 6H). The ^13^C NMR (101 MHz, CDCl_3_) δ 208.1, 139.9, 128.6, 128.3, 126.0, 70.1, 70.0, 62.9, 43.9, 43.8, 36.3, 31.1, 25.8, 25.8, 18.3, 18.0, −3.5, −4.3, −4.9, −5.4. HRMS-ESI (*m*/*z*): [M + H]^+^ calcd. for C_26_H_47_NO_3_Si_2_, 448.4378, found 448.4382. Copies of ^1^H and ^13^C NMR could be found in Appendix A.

#### (5-(((*tert*-Butyldimethylsilyl)oxy)methyl)-1-phenethyl-1*H*-pyrrol-2-yl)methyl acetate (**16**)

To a stirred solution of Octamethyl-8-(1-phenethylaziridin-2-yl)-4,9-dioxa-3,10-disiladodecan-6-one **15** (1.5 g, 3.13 mmol) in dry CH_2_Cl_2_ (30 mL) was added acetic acid (0.56 mL, 6.27 mmol) at 0 °C under an N_2_ atmosphere. After 6 h stirred at 0 °C, the reaction mixture was quenched with saturated aqueous NH_2_CO_3_ (10 mL). The organic layer was separated, and the aqueous layer was extracted with CH_2_Cl_2_ (2 × 20 mL). The organic layer was dried over Na_2_SO_4_ and concentrated in vacuo to obtain the crude product, which was purified by column chromatography (EtOAc/hexanes, 2:8) to give pure (5-(((*tert*-butyldimethylsilyl)oxy)methyl)-1-phenethyl-1*H*-pyrrol-2-yl)methyl acetate **16** as a yellow liquid (1.0 g, 82%) yield. The ^1^H NMR (400 MHz, CDCl_3_) δ 7.30 (dd, *J* = 7.9, 6.4 Hz, 2H), 7.23 (d, *J* = 7.4 Hz, 1H), 7.14–7.11 (m, 2H), 6.15 (d, *J* = 3.5 Hz, 1H), 6.00 (d, *J* = 3.5 Hz, 1H), 4.96 (s, 2H), 4.53 (s, 2H), 4.17 (t, *J* = 6.5 Hz, 2H), 3.06 (t, *J* = 6.2 Hz, 2H), 2.06 (s, 3H), 0.89 (s, 9H), 0.05 (s, 6H). The ^13^C NMR (101 MHz, CDCl_3_) δ 170.7, 138.6, 133.5, 128.8, 128.6, 127.3, 126.6, 110.4, 108.0, 57.9, 57.6, 45.8, 38.0, 25.9, 21.1, 18.3, −5.2. HRMS-ESI (*m*/*z*): [M + Na]^+^ calcd. for C_22_H_33_NNaO_3_Si, 410.6150, found 410.6158. Copies of ^1^H and ^13^C NMR could be found in Appendix A.

#### (5-(((*tert*-Butyldimethylsilyl)oxy)methyl)-1-phenethyl-1*H*-pyrrol-2-yl)methanol (**17**)

To a stirred solution of (5-(((*tert*-butyldimethylsilyl)oxy)methyl)-1-phenethyl-1H-pyrrol-2-yl)methyl acetate **16** (0.7 g, 1.80 mmol) in MeOH (10 mL) was added potassium carbonate (0.249 g, 1.80 mmol) at 0 °C, and the mixture was stirred for 1 h at rt. Then, MeOH was removed under vacuum and extracted with CH_2_Cl_2_ (2 × 10 mL). The organic layer was dried over Na_2_SO_4_ and concentrated in vacuo to obtain the crude product, which was purified by column chromatography (EtOAc/hexanes, 4:6) to give pure (5-(((*tert*-butyldimethylsilyl)oxy)methyl)-1-phenethyl-1H-pyrrol-2-yl)methanol (**17**) as a yellow liquid (0.5 g, 80% yield). The ^1^H NMR (400 MHz, CDCl_3_) δ 7.32–7.20 (m, 3H), 7.15–7.10 (m, 2H), 6.01 (d, *J* = 3.5 Hz, 1H), 5.97 (d, *J* = 3.5 Hz, 1H), 4.55 (s, 2H), 4.42 (s, 2H), 4.24 (t, *J* = 6.5 Hz, 2H), 3.10 (t, *J* = 6.2 Hz, 2H), 0.90 (s, 9H), 0.06 (s, 6H). The ^13^C NMR (101 MHz, CDCl_3_) δ 138.9, 133.1, 132.7, 128.9, 128.5, 126.5, 107.8, 107.7, 57.6, 56.9, 45.7, 38.0, 25.9, 18.3, −5.2. HRMS-ESI (*m*/*z*): [M + H]^+^ calcd. for C_20_H_32_NO_2_Si, 346.5226, found 346.5231.

#### 5-(((*tert*-Butyldimethylsilyl)oxy)methyl)-1-phenethyl-1*H*-pyrrole-2-carbaldehyde (**18**)

To a stirred solution of alcohol **17** (0.5 g, 1.29 mmol) in dry CH_2_Cl_2_ (4 mL) was added Dess–Martin periodinane (0.820 g, 1.93 mmol) at 0 °C, and the mixture was stirred for 1 h at rt. Then, the reaction mixture was quenched with a 1:1 mixture of saturated solution of NaHCO_3_ (10 mL) and extracted with CH_2_Cl_2_ (2 × 10 mL). The organic layer was dried over Na_2_SO_4_ and concentrated in vacuo to obtain the crude product, which was purified by column chromatography (EtOAc/hexanes, 2:8) to give pure aldehyde **18** as a yellow liquid (330 mg, 74% yield). The ^1^H NMR (400 MHz, CDCl_3_) δ 9.55 (s, 1H), 7.27 (dd, *J* = 5.2, 2.1 Hz, 3H), 7.16–7.13 (m, 2H), 6.90 (d, *J* = 4.0 Hz, 1H), 6.11 (d, *J* = 4.0 Hz, 1H), 4.53 (t, *J* = 6.5 Hz, 2H), 4.31 (s, 2H), 3.04 (t, *J* = 6.2 Hz, 2H), 0.89 (s, 9H), 0.04 (s, 6H). The ^13^C NMR (101 MHz, CDCl_3_) δ 179.2, 142.2, 138.6, 132.0, 129.0, 128.4, 126.5, 124.5, 109.6, 57.0, 47.6, 37.6, 25.8, 18.2, −5.3. HRMS-ESI (*m*/*z*): [M + H]^+^ calcd. for C_20_H_30_NO_2_Si, 344.2264, found 344.2269. Copies of ^1^H and ^13^C NMR could be found in Appendix A.

#### 5-(Hydroxymethyl)-1-phenethyl-1*H*-pyrrole-2-carbaldehyde (**19**)

To a stirred solution of 5-(((*tert*-butyldimethylsilyl)oxy)methyl)-1-phenethyl-1H-pyrrole-2-carbaldehyde (**18**) (0.3 g, 0.87 mmol) in dry THF (10 mL) was added TBAF (0.94 mL, 1.0 M in THF, 0.96 mmol) at 0 °C and stirred for 1 h. After completion of the reaction was quenched with saturated aqueous NH_2_CO_3_ (10 mL). The organic layer was separated, and the aqueous layer was extracted with ethyl acetate (2 × 20 mL). The organic layer was dried over Na_2_SO_4_ and concentrated in vacuo to obtain the crude product, which was purified by column chromatography (EtOAc/hexanes, 3:7) to give pure 5-(hydroxymethyl)-1-phenethyl-1H-pyrrole-2-carbaldehyde **19** as a yellow oil (168 mg, 84% yield). The ^1^H NMR (400 MHz, CDCl_3_) δ 9.58 (s, 1H), 7.27–7.21 (m, 3H), 7.10 (d, *J* = 6.5 Hz, 2H), 6.93 (d, *J* = 4.0 Hz, 1H), 6.17 (d, *J* = 4.0 Hz, 1H), 4.55 (t, *J* = 7.2 Hz, 2H), 4.29 (s, 2H), 3.05 (t, *J* = 7.2 Hz, 2H). The ^13^C NMR (101 MHz, CDCl_3_) δ 179.4, 141.7, 138.5, 132.2, 129.0, 128.6, 126.7, 124.6, 110.0, 56.3, 47.6, 37.7. HRMS-ESI (*m*/*z*): [M + H]^+^ calcd. for C_14_H_16_NO_2_, 230.1178, found 230.1185. Copies of ^1^H and ^13^C NMR could be found in Appendix A.

## 4. Conclusions

In summary, multi-substituted pyrroles were synthesized from regiospecific aziridine ring opening and subsequently intramolecular cyclization with a carbonyl group at the *γ*-position in the presence of Lewis acid (TMSN_3_ or ZnCl_2_) or protic acid (AcOH). This method is high atom economical in that all reactants are incorporated into the final product with the removal of water. This new protocol can be applied to the synthesis of various pyrroles, including natural products.

## Data Availability

Not applicable.

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
