# Peer review of "Atom Economical Multi-Substituted Pyrrole Synthesis from Aziridine"

_molecules, 2022, doi:10.3390/molecules27206869_

Round 1

Reviewer 1 Report

This manuscript describes the Pyrrole Synthesis from aziridines. A variety of substituted pyrroles were obtained. These heterocycles are important structures in biological and pharmaceutical fields. Further, natural product is synthesized. Products are good yields and their structure are well characterized. From these reasons, I recommend this manuscript for publication, provided that the following issues are addressed.

1)   Line 50: e most?

2)   Scheme 1,2 and 5 eqs: text styles are strange.

3)    It is better that the results using TMSCl and TMSCN in Table 1 is discussed in Scheme 2.

4)    In Scheme 2 for 2i-2k, it is better that using ZnCl2 catalyst is indicated.

5)    Line 94 and 112: alpha-position, alpha oxonium

6)    Line 100: pyrrole 2 is bold

7)    Scheme 5: -H2O is added under arrow of 7 to 8.

8)    Line 114-117 and Scheme 5: Oxidative removal of hydride of 8 gives 10 not 9. Deprotonation is from 10 to 4 not 9 to 10. Intermediate 9 is not necessary.

Author Response

RESPONSE TO REVIEWER’S COMMENTS:

Reviewer 1:

This manuscript describes the Pyrrole Synthesis from aziridines. A variety of substituted pyrroles were obtained. These heterocycles are important structures in biological and pharmaceutical fields. Further, natural product is synthesized. Products are good yields and their structure are well characterized. From these reasons, I recommend this manuscript for publication, provided that the following issues are addressed.

Q1:   Line 50: e most?

Ans: As suggested by reviewer, typographical error “e” has been removed in the manuscript.

Q2:   Scheme 1,2 and 5 eqs: text styles are strange.

Ans: As suggested by reviewer, we changed text styles, structures and reagents in schemes 1, 2 & 5 eqs.

Q3:    It is better that the results using TMSCl and TMSCN in Table 1 is discussed in Scheme 2.

Ans: As suggested by the reviewer, we removed TMSCl and TMSCN in table 1 and incorporated the results of TMSCl and TMSCN and discussed in Scheme 2.

Q4.    In Scheme 2 for 2i-2k, it is better than using the ZnCl2 catalyst is indicated.

Ans1: Yes, we changed as you recommended.

Q5:    Line 94 and 112: alpha-position, alpha oxonium

Ans1: Yes, we changed as you recommended.

Q6:    Line 100: pyrrole 2 is bold

Ans1: Yes, we changed as you recommended.

Q7:   Scheme 5: -H2O is added under arrow of 7 to 8.

Ans1: Yes, we changed as you recommended.

Q8:   Line 114-117 and Scheme 5: Oxidative removal of hydride of 8 gives 10 not 9. Deprotonation is from 10 to 4 not 9 to 10. Intermediate 9 is not necessary.

Ans: As suggested by the reviewer, we revised the mechanism and chemical structures and description in the text related (Line 114-117) to Scheme 5.

Reviewer 2 Report

The work by H-J. Ha et al is about the preparation of pyrroles. The procedure can be useful for the scientific community. It can be published after changes detailed below are done:

-In Scheme 1 reaction (4) solvent in the arrow and legend of substituents in the product are not displayed well.

-In the arrow reaction of Table 1 "N3-" instead of Nu should be indicated.

-In page 4 last paragraph says "Dess-Martin periodate" but should say "Dess-Martin periodinane". Also, from my view the paragraph should be re-written, the first  word "Surprisingly" should be removed. Then instead of saying "a complex intermediate, which was reacted further for ring- opening with various nucleophiles like OMe, OAc, Cl, and CN without purification to high yields of 2,3-disubstituted pyrrole 5-aldehydes " should say "a complex mixture of compounds which were directly reacted for ring- opening with various nucleophiles like OMe, OAc, Cl, and CN  to afford 2,3-disubstituted pyrrole 5-aldehydes in one-pot procedure".

-IN Scheme 3, yields intervals should be indicated for both reactions conducting to 2 or 4.

-The legend of "Scheme 4. Examples of synthesis of 2-formyl pyrroles starting from. " should be completed.

-In Scheme 5 says "Paar-Knorr reaction" but should say "Paal-Knorr reaction". Some legends and reagents do not look fine. 

-In the mechanism of Scheme 5, the authors go to great lengths trying to explain the chemical transformation which is just: opening of aziridine and Paul_Knorr. The below mechanism is wrong. Basically the above mechanism pathway (from 6 to 2 or 4) is fine (Paal-Knorr) and then to explain the aldehyde formation is an oxidation of the primary alcohol with the oxidant (Dels-Martin). 

-There are two Scheme 5, second should be indicated as Scheme 6

-The whole "Scheme 6" (chemical structures, reagents, conditions,...) do not look fine.

-In "Scheme 6" the step from 13 to 14 says TBSCl but in the text says TBSOTf.

-At the end of first paragraph of page 6 says (Scheme 5) but should say (Scheme 6).

-In the Experimental Section, the procedure for the preparation compounds 4 using  Dess-Martin reagent is not described.

Author Response

Reviewer 2:

The work by H-J. Ha et al is about the preparation of pyrroles. The procedure can be useful for the scientific community. It can be published after changes detailed below are done:

Q1: In Scheme 1 reaction (4) solvent in the arrow and legend of substituents in the product are not displayed well.

Ans: Yes, we changed as you recommended to make it clearer.

Q2. In the arrow reaction of Table 1 "N3-" instead of Nu should be indicated.

Ans: Yes, we changed as you suggested.

Q3. In page 4 last paragraph says "Dess-Martin periodate" but should say "Dess-Martin periodinane". Also, from my view the paragraph should be re-written, the first word "Surprisingly" should be removed. Then instead of saying "a complex intermediate, which was reacted further for ring- opening with various nucleophiles like OMe, OAc, Cl, and CN without purification to high yields of 2,3-disubstituted pyrrole 5-aldehydes " should say "a complex mixture of compounds which were directly reacted for ring- opening with various nucleophiles like OMe, OAc, Cl, and CN to afford 2,3-disubstituted pyrrole 5-aldehydes in one-pot procedure".

Ans: Yes, we changed “Next, oxidation of secondary alcohol of compound 3 at the γ-position of aziridine with Dess–Martin periodinane in CH2Cl2 yielded a complex mixture of compounds which were directly reacted for the ring- opening with various nucleophiles like OMe, OAc, Cl, and CN to afford 2,3-disubstituted pyrrole 5-aldehydes (4a-4d) in the one-pot procedure as shown in Scheme 3 with examples in the Scheme 4”.

Q4. In Scheme 3, yield intervals should be indicated for both reactions conducting to 2 or 4.

Ans: Yes, we indicated yield intervals for both reactions conducting to 2 or 4 in Scheme 3, as you recommended.

Q5. The legend of "Scheme 4. Examples of synthesis of 2-formyl pyrroles start from. "Should be completed.

Ans: Yes, we removed the typographical error in the sentence “starting from” as you recommended.

Q6. In Scheme 5 says "Paar-Knorr reaction" but should say "Paal-Knorr reaction". Some legends and reagents do not look fine. 

Ans: As suggested by the reviewer, we revised the mechanism and chemical structures and description in the text related to Scheme 5.

Q7. In the mechanism of Scheme 5, the authors go to great lengths trying to explain the chemical transformation which is just: opening of aziridine and Paul Knorr. The below mechanism is wrong. Basically, the above mechanism pathway (from 6 to 2 or 4) is fine (Paal-Knorr) and then to explain the aldehyde formation is an oxidation of the primary alcohol with the oxidant (Dess-Martin). 

Ans: As suggested by the reviewer, we revised the mechanism and chemical structures and description in the text related to Scheme 5.

Q8. There are two Scheme 5, the second should be indicated as Scheme 6.

Ans: Yes, we changed as you recommended.

Q9. The whole "Scheme 6" (chemical structures, reagents, conditions) do not look fine.

Ans: Yes, we corrected the whole "Scheme 6" (chemical structures, reagents, conditions) as you recommended.

Q10. In "Scheme 6" the step from 13 to 14 says TBSCl but in the text says TBSOTf.

Ans: Yes, we corrected as “TBSOTf” instead of “TBSCl” in Scheme 6.

Q11. At the end of first paragraph of page 6 says (Scheme 5) but should say (Scheme 6).

Ans: Yes, we changed as you recommended.

Q12. In the Experimental Section, the procedure for the preparation compounds 4 using Dess-Martin reagent is not described.

Ans: Yes, we incorporated the procedure for the preparation compounds 4a using Dess-Martin reagent and followed by Lewis’s acid mediated aziridine ring opening by OMe Nucleophile from compound 3 in experimental section.

Reviewer 3 Report

1. Electron pushing arrows in intermediate 9 is not convincing, should be redrawn in other way.

2. In page 2, “2+3” cycloaddition should be replaced by [3+2]

3. In page 2, check this wording “First, e most of the methods” -not clear

4. Scheme 5 repeated, last scheme should be “scheme 6”.

5. Extensive corrections to figures and revision of the overall structures will improve the quality of manuscript. For example, Scheme1; eq 4, Table1 foot note, scheme 2, scheme 5 and scheme 6 figures are not written in a proper way, or it may happen while convert word file to pdf. Please correct them all.

6. In scheme 5, at any time point, authors have isolated or identified by mass of intermediate 5…?

Author Response

Reviewer 3:

Q1. Electron pushing arrows in intermediate 9 is not convincing, should be redrawn in other way.

Ans: As suggested by the reviewer, we revised the mechanism and chemical structures and description in the text related to Scheme 5.

Q2. In page 2, “2+3” cycloaddition should be replaced by [3+2]

Ans: Yes, we changed as you recommended.

Q3. In page 2, check this wording “First, e most of the methods” -not clear

Ans: Yes, we corrected a typographical error “e most” in page 2.

Q4. Scheme 5 repeated, last scheme should be “scheme 6”.

Ans: Yes, we changed as you recommended.

Q5. Extensive corrections to figures and revision of the overall structures will improve the quality of manuscript. For example, Scheme1; eq 4, Table1 foot note, scheme 2, scheme 5 and scheme 6 figures are not written in a proper way, or it may happen while convert word file to pdf. Please correct them all.

Ans: Yes, we changed figures and structures in Scheme1; eq 4, Table1 foot note, scheme 2, scheme 5 and scheme 6 as you recommended.

Q6. In scheme 5, at any time point, authors have isolated or identified by mass of intermediate 5.?.

Ans: Thank you for your suggestion. However, it is not possible to isolate or observe any intermediate due to its fast reaction processes and unstable intermediates.